# Diagnostic Accuracy of Contrast-Enhanced, Spectral Mammography (CESM) and 3T Magnetic Resonance Compared to Full-Field Digital Mammography plus Ultrasound in Breast Lesions: Results of a (Pilot) Open-Label, Single-Centre Prospective Study

**DOI:** 10.3390/cancers14051351

**Published:** 2022-03-07

**Authors:** Francesca Romana Ferranti, Federica Vasselli, Maddalena Barba, Francesca Sperati, Irene Terrenato, Franco Graziano, Patrizia Vici, Claudio Botti, Antonello Vidiri

**Affiliations:** 1Radiology and Diagnostic Imaging Department, IRCCS Regina Elena National Cancer Institute, Via Elio Chianesi 53, 00144 Rome, Italy; francescaromana.ferranti@ifo.it (F.R.F.); federica.vasselli@gmail.com (F.V.); 2Division of Medical Oncology 2, IRCCS Regina Elena National Cancer Institute, Via Elio Chianesi 53, 00144 Rome, Italy; 3Biostatistics-Scientific Direction, IRCCS Regina Elena National Cancer Institute, Via Elio Chianesi 53, 00144 Rome, Italy; francesca.sperati@ifo.it (F.S.); irene.terrenato@ifo.it (I.T.); 4Division of Breast Surgery, IRCCS Regina Elena National Cancer Institute, Via Elio Chianesi 53, 00144 Rome, Italy; franco.graziano@ifo.it (F.G.); claudio.botti@ifo.it (C.B.); 5Sperimentazioni di Fase IV, IRCCS Regina Elena National Cancer Institute, 00144 Rome, Italy; patrizia.vici@ifo.it

**Keywords:** breast cancer, MRI, CESM, breast imaging

## Abstract

**Simple Summary:**

The characterization of breast lesions by magnetic resonance imaging (MRI) is influenced by a high rate of false positives. Contrast-enhanced spectral mammography (CESM) is a promising modality that seems to compensate for the high costs, times and main limitations of MRI. The aim of our prospective study was to evaluate the diagnostic accuracy of CESM in comparison to 3T MRI imaging in the characterization of breast lesions. We enrolled 118 patients and histologically assessed 142 breast lesions. Patients underwent full-field digital mammography (FFDM), ultrasound (US), with CESM and MRI assessing the diagnostic accuracy of CESM. Sensitivity (Se), specificity (Sp), positive predictive value (PPV) and negative predictive value (NPV) were considered as measures of accuracy of different diagnostic procedures in predicting the nature and characteristics of the examined breast lesions.

**Abstract:**

Introduction: To assess the diagnostic accuracy of CESM and 3T MRI compared to full-field digital mammography (FFDM), plus US, in the evaluation of advanced breast lesions. Materials and Methods: Consenting women with suspicious findings underwent FFDM, US, CESM and 3T MRI. Breast lesions were histologically assessed, with histology being the gold standard. Two experienced breast radiologists, blinded to cancer status, read the images. Diagnostic accuracy of (1) CESM as an adjunct to FFDM and US, and (2) 3T MRI as an adjunct to CESM compared to FFDM and US, was assessed. Measures of accuracy were sensitivity (Se), specificity (Sp), positive predictive value (PPV) and negative predictive value (NPV). Results: There were 118 patients included along with 142 histologically characterized lesions. K agreement values were 0.69, 0.68, 0.63 and 0.56 for concordance between the gold standard and FFDM, FFDM + US, CESM and MRI, respectively (*p* < 0.001, for all). K concordance for CESM was 0.81 with FFDM + US and 0.73 with MRI (*p* value < 0.001 for all). Conclusions: CESM may represent a valuable alternative and/or an integrating technique to MRI in the evaluation of breast cancer patients.

## 1. Introduction

Female breast cancer is the most common cancer diagnosed worldwide, ranking fifth among the most frequent causes of cancer death [1]. In recent years, relevant advances in breast cancer diagnostics have been achieved. Contrast-enhanced dual-energy spectral mammography (CESM) is an imaging technique based on the use of a dual-energy approach for cancer detection. In more detail, with CESM, the low-energy (LE) component of breast images renders the morphological information similar to a two-dimensional (2D) digital mammography, while the high-energy (HE) component of breast images demonstrates post-contrast enhanced mammograms by using the K-edge effect of iodine, allowing for the evaluation of tumor neovascularity [2]. Although CESM is still in the early stages of clinical use, results from the related studies appear extremely encouraging [3]. When evaluating 143 breasts of 72 women and considering histopathology to be the gold standard, Mori and colleagues observed significantly greater sensitivity, specificity and accuracy of CESM compared to conventional full-field digital mammography (FFDM) [4].

The findings concerning CESM sensitivity have been strengthened by the results of the systematic review and meta-analysis of eight eligible studies that included 920 patients with 994 lesions [5]. In addition, in a feasibility study of CESM compared to FFDM to assess residual tumors following neoadjuvant chemotherapy, the sensitivity and specificity of CESM were about 83% and 100%, respectively, compared to 50% (for both) with FFDM. Similarly, positive predictive and negative predictive values were higher for CESM [6]. Adjunctive evidence in support of the superior accuracy of CESM in evaluating tumor size compared to FFDM and ultrasound (US), both in dense and non-dense breasts, is derived from a recent study [7]. Finally, CESM has shown increased diagnostic performance compared to FFDM in lower-prevalence patient populations, such as those referred from breast cancer screening [8]. However, exclusive evidence from an extremely recent systematic review and meta-analysis, including prospective trials, was not supportive. The accuracy of CESM is now under debate [9].

MRI is an extremely accurate imaging tool in breast cancer diagnosis. Breast MRI can efficiently integrate diagnostic information to identify breast lesions not definitively diagnosed by FFDM and US. MRI sensitivity in cancer diagnosis has been estimated as ranging from 80 to 97.8%, while specificity is between 46 and 93.3%. The high rate of false positives, along with the high expenses, difficult accessibility and contraindications in patients with metallic implants and pacemakers, poses remarkable limitations to its use in breast cancer diagnostics and screening programs [10,11].

Globally considered, the above-cited literature fueled our interest in this timely topic, which culminated in the design and performance of a pilot, open-label, prospective study conceived and conducted at our facility, the Regina Elena National Cancer Institute. Our trial aimed to compare diagnostic accuracy of CESM and 3T MRI with FFDM and US to assess breast cancer lesions. In more detail, we evaluated diagnostic accuracy of CESM as adjunct to FFDM + US and MRI as adjunct to FFDM + US. In doing so, we addressed an extremely timely issue to a research agenda focused on the assessment and characterization of breast lesions.

## 2. Materials and Methods

This study was conceived as an open-label, single-center study carried out at the Radiology Department of the Regina Elena National Cancer Institute. The study protocol and inherent consent form were submitted to the Institutional Review Board for formal evaluation and then approval (RS 410/13). The outcome assessment was based on the evaluation of accuracy expressed in terms of sensitivity (Se), specificity (Sp), positive predictive value (PPV) and negative predictive value (NPV). Data analysis was also performed using SPSS version 20.0 (SPSS Inc., Chicago, IL, USA).

Consenting women with unresolved/suspicious findings after FFDM and US underwent a diagnostic workup that included baseline (FFDM and US) reassessment, plus CESM and 3T MRI after being informed and after having signed an informed consent form. The timing of breast imaging differed by menopausal status. In postmenopausal women, defined as a self-reported (at least) 12-month period of amenorrhea, the estimated time interval between baseline reassessment and performance of CESM was (on average) 1 week. This same time window was expected between CESM and administration of the paramagnetic contrast agent (gadolinium) and between the latter, as well as 3T MRI. In premenopausal women, any of the aforementioned procedures were performed during the second week of their menstrual cycle. The actual nature and characterization of breast lesions was histologically assessed in samples collected by core biopsy and/or surgery. Two experienced breast radiologists read the FFDM, US, CESM and 3T MRI, with a minimum 4-week washout period among different sessions for the same patient. Outcome assessors were blinded to cancer status and were able to work in consensus.

Any breast lesions detected were characterized based on data gathered from breast imaging and pathology records. Features of interest included: lesion presence and location (breast site and quadrant); finding the type (e.g., mass and architectural distortion) and size (largest diameter) from FFDM and US, CESM and 3T MRI; BIRADS classification (FFDM, CESM and 3T MRI) (12); mammographic density (FFDM); type of contrast kinetic curves (CESM and 3T MRI); and (pathologic) diagnosis assessed by cytology/histology (e.g., benign/malignant).

Patients of at least 18 years of age were eligible if presenting with unresolved/suspicious findings after FFDM plus US, e.g., breast mass, breast asymmetry and distortion. Moreover, fine-needle aspiration biopsy, core-needle biopsy and/or breast surgery had to be scheduled independently for our study purposes. Written informed consent was secured from each participating patient. Eligibility was not granted if one or more of the following was verified: prior diagnosis of breast and/or other malignancy/ies, except adequately treated non-melanoma skin cancer and/or curatively treated in situ cancer of the cervix; exclusive evidence of clusters of microcalcifications in the absence of previously mentioned findings (e.g., mass and asymmetry); history of allergy to contrast medium and/or multiple allergies; pregnancy and lactation; and breast implants or any unstable medical conditions, which could interfere with safe participation in the trial. All eligible participants repeated baseline FFDM and US in radiology. Each patient underwent CESM and 3T MRI. A detailed description of the procedures and related protocols is reported below.

### 2.1. CESM Examinations

CESM examinations were performed with a device developed by GE Healthcare (Chicago, IL, USA), allowing dual-energy CESM acquisitions. It uses a current full-field digital mammography system, SenoBright, or Senographe Essential. A catheter was inserted into the antecubital vein of the contralateral arm to the breast of concern. A one-shot intravenous injection of 1.5 mL/body weighted contrast agent (Visipaque 320, GE, Oslo, Norvegia) was then performed using a power injector. Two minutes after injection of the contrast agent, a bilateral mammography examination with standard compression was performed in craniocaudal (CC) and medio-lateral-oblique (MLO) views, each with a pair of low- and high-energy exposures and in a total examination time of 6 min. A combination of low-energy and high-energy images through specific image processing was automatically provided by the system to generate subtracted images with contrast agent uptake information (one in the MLO and one in the CC view of each breast). The mean, minimum and maximum X-ray doses administered in the course of FFDM and CESM are shown in Table 1.

### 2.2. T MR Examinations

Breast MRI scan was conducted with 3T Tomography (Discovery MR750w GE Medical Systems, Waukesha, WI, USA) and with an 8-channel surface coil to assess the patients’ bilateral breasts in a prone position.

For a morphological contrast evaluation, a T2-weighted fast spin-echo axial FSE was used (slice thickness 3 mm; gap interslice 0.3 mm; matrix 512 × 512; FOV 32–40 cm). Short time inversion recovery axial STIR sequence suppressed the adipose tissue (slice thickness 3 mm; gap interslice 0.3 mm; matrix 512 × 512; FOV 32–40 cm). The dynamic contrast evaluation was based on an axial 3D dynamic T1 sequence, weighted by 1 basal acquisition and 5 post-contrast acquisitions of Gd-DTPA of 0.2 mmol/kg at a speed of 2 mL/s followed by 20 mL of saline solution. The 3T MR examination might include a sagittal 3D FSPGR sequence with fat-saturated imaging, post-contrast injection on the suspicious breast with shimming delimitation (slice thickness 1 mm; gap interslices 0 mm; matrix 360 × 352; FOV 32–40 cm). Diffusion imaging was performed on a single-shot axial echo-planar imaging (EPI) with a diffusion gradient amplitude on 3 octagonal axels and b-values of 0 and 800 s/mm (slice thickness 4 mm; gap interslices 0 mm; matrix 128 × 128; FOV 32–40 cm)

### 2.3. Image Analysis for FFDM, US and CESM

Baseline FFDM and US examinations for the reading session were repeated on study entrance. For the two radiologists, readings were performed on individual workstations, loaded with all cases, and calibrated in controlled ambient lighting conditions. Prior to studying the readings, a training session was held to familiarize radiologists with CESM and the reading protocol. Iodine-enhanced CESM images were reviewed by criteria of the American College of Radiology, Reston, VA, USA [12].

To reduce bias from image recall, for each study participant, readings occurred in two different sessions separated by a 1-month washout interval. Thus, for the same patient, the first reading session included images from FFDM + US and 3T MRI, whereas CESM-related images were evaluated in the second reading session. In both sessions, the two radiologists assessed the case, localized findings, assigned BI-RADS scores and completed and saved an electronic data form.

This form included patient identification, breast density on FFDM (BI-RADS scores of 1 to 4), location of each finding, type of finding (e.g., asymmetry, mass, scar/distortion), degree of confidence in the presence of each finding (5-step scale in which 1 = very low and 5 = very high), BI-RADS classification (1 to 5, BI-RADS scores of 0 and 6 not allowed). The electronic assessment was completed following the histological characterization of samples obtained by biopsy/surgery.

### 2.4. Statistical Analysis

Descriptive statistics were computed for all variables of interest. Continuous variables were reported as means and standard deviations, while categorical variables were reported as frequencies and percentage values. Sensitivity (Se), Sp, PPV and NPV were chosen to report on the accuracy of the diagnostic procedures with the scope to predict the nature and characteristics of the breast lesions under examination.

Lesion characteristics (lesion presence and location, finding type and size) on images obtained throughout the techniques were considered for potential associations with pathologic features, as assessed by cytology/histology.

The agreement between diagnostic tests and the gold standard, or CESM, was estimated by using raw agreement and the Cohen’s kappa statistics [13], interpreted with the Landis and Koch classification criteria [14]. Inter-reader variability was reported in terms of kappa statistics. The inter-observer agreement, diagnostic accuracy (DA), sensitivity (Se), specificity (Sp), positive predictive value (PPV) and negative predictive value (NPV) were calculated for all the imaging techniques included in our study. A *p*-value < 0.05 was considered statistically significant. All statistical analyses were performed with SPSS statistical software version 20 (SPSS Inc., Chicago, IL, USA).

## 3. Results

Overall, 118 female patients contributed data to our analysis. As per the study protocol, all patients were assessed by FFDM + US and MRI + CESM. Patient characteristics are shown in Table 2. Mean age at study entry was 48.5 (±9.7), with 79 of 118 patients menopausal. In 88 (74.6%) patients, breast histology revealed the neoplastic nature of the lesion/s assessed. Seven patients had the nature of the lesion assessed by cytology, of which three with malignant results were addressed by biopsy. In 70 (59.3%) patients, we found evidence of monofocal lesions (Figure 1), while 18 (20.5%) showed multifocal breast cancer (Figure 2), and 8 (6.8%) had bilateral breast cancer. In five (4.2%) patients, we only assessed benign lesions. Conversely, in 24 patients (20.3%), one benign lesion could coexist with monofocal breast cancer, but in 37 patients (31.4%), a benign lesion was associated with multifocal or bilateral breast cancer. A total of 57 of 88 patients underwent mastectomy, and 11, quadrantectomy.

Breast lesions’ characteristics are reported in Table 3. BI-RADS descriptors for each mode of assessment are listed as well. We observed 105 lesions with FFDM, 137 lesions with US, 110 lesions with CESM and 108 lesions with 3T MRI. The total number of lesions assessed by cytology, biopsy or surgery was 142. In 88 of 118 patients, breast cancer was ultimately diagnosed. In 62 patients (52.5%), we observed unifocal breast cancers, while in 18 (15.2%), we observed multifocal breast cancers, and in 8 (6.8), bilateral breast cancers were found (results reported in Table 2).

The results of concordance between CESM and 3T MR and the gold standard were encouraging. Table 4 summarizes inherent findings based on a “per lesion” analysis. Kappa values were 0.69, 0.81, 0.76 and 0.68 for concordance between FFDM and the gold standard, FFDM and CESM, and FFDM and US. CESM and 3T MRI showed a K concordance with the gold standard of 0.63 and 0.56, respectively. K concordance between CESM and the other modalities showed the following results: 0.81 with FFDM + US, 0.73 with MRI; *p* value ≤ 0.001 for all reported results. These same tests were performed according to a “per patient” approach analysis, showing a slightly lower K concordance between the gold standard, CESM and MRI (0.52 and 0.37, with a *p* value equal to 0.03 and 0.018, respectively) in monofocal breast cancers and a K concordance of 0.44 with a *p* value of 0.057 between MRI and the gold standard for multifocal and bilateral breast cancers. These results are extensively reported in Table 5.

Results on the diagnostic accuracy are reported in Table 6. Sensitivity (Se) was 0.99 for FFDM, 0.97 for US, 0.98 for the combination of FFDM + US, 0.99 for MRI and 1 for CESM. Specificity (Sp) was 0.61 for FFDM, 0.76 for US, 0.63 for the combination of FFDM + US, 0.47 for MRI and 0.5 for CESM. The PPV was 0.94 for FFDM, 0.92 for US, 0.89 for the combination of FFDM + US, 0.88 for MRI and 0.92 for CESM. The NPV was 0.89 for FFDM, 0.88 for US, 0.90 for the combination of FFDM + US, 0.9 for MRI and 1 for CESM. The DA was 0.93 for FFDM, 0.92 for US, 0.89 for the combination of FFDM + US, 0.88 for MRI and 0.93 for CESM. Results on the inter-observer agreement are shown in Table 7. The K values were 0.53 for FFDM, and 1.0 for US, MRI and CESM, respectively.

## 4. Discussion

CESM is increasingly emerging as an imaging technique with a key role in the diagnostic workup of patients affected by breast cancer [2]. When compared to MRI, some major strengths of CESM are represented by faster performance modality, lower costs and better tolerance, particularly in patients with physical limitations and/or claustrophobia [15]. The increasing interest toward CESM, with some encouraging data in the literature, prompted us to perform an open-label, single-center study of 118 patients in need of a diagnostic workup.

According to our study protocol, all patients enrolled underwent baseline assessment throughout FFDM + US. Subsequently, breast lesions were also studied by CESM and MRI, with cytological/histological assessment being the gold standard. Correlation analysis was performed according to a *per patient* and *per lesion* approach. Overall, CESM showed good concordance compared to the gold standard and the other modalities. When analyzing our data according to a *per lesion* approach, MRI, though maintaining a significant *p* value, tended to show a slightly lower K concordance. This aspect is evident in the *per patient* analysis, while concordance is not even significant when evaluating multifocal and bilateral breast cancers. This lack of concordance on MRI can be at least partly justified by the limits of this modality in terms of low specificity, which translates into a high rate of false positive cases [16]. Our results suggest that CESM could provide a compensation to MRI by means of a minor propensity to false positive cases. This aspect has been widely studied, as even recent work confirms encouraging results in terms of better specificity rates of CESM compared to MRI [17]. In a further attempt to improve specificity, an interesting study investigated the feasibility of CESM in combination with molecular imaging in the pre-surgical evaluation window. Data showed similar visualization of index cancers, with higher specificity for CESM compared to MRI. However, undergoing multimodal pre-surgical assessment was associated with discomfort in a relevant number of patients [18]. 

Patients undergoing CESM are exposed to radiation. However, recent literature shows a trend toward dose reduction, including those undergoing tomosynthesis. This would undoubtedly add to CESM advantages in terms of cost, time and patient tolerance [19,20].

The study herein presented does not include investigational tasks related to the combined use of digital breast tomosynthesis (DBT) and CESM. Indeed, the use of BDT in breast cancer diagnosis has recently grown, along with the detection rate of architecture distortion (AD). CESM may provide an effective tool in the evaluation of tomosynthesis-detected architectural distortion (AD). According to recently published data, the absence of AD lesion enhancement on CESM against a background of minimal or mild background parenchymal enhancement may indicate a lower risk of breast malignancy. The negative NPV reported by the authors is remarkably high, i.e., 100% in 26 (27.7%) of the 94 lesions with no suspicious enhancement on CESM, and encourages further investigation. It is our firm intention to include the use of digital breast tomosynthesis in combination with CESM in future studies within this same research pipeline [21].

Among the hardest and most intriguing challenges of CESM, its use in women with dense breasts deserves mentioning. In a pilot study of 318 women at increased breast cancer risk, CESM and MRI were both able to detect breast cancers not previously highlighted on conventional mammography. Neither did these two techniques differ by positive predictive value and specificity to a significant extent. However, given the still limited evidence, the authors carefully encourage the consideration of CESM in the evaluation of women at increased breast cancer risk who do not meet the criteria for MRI or for whom access to MRI is limited [22].

Results on DA, Se, Sp, PPV and NPV are consistent with those from the available literature [23]. We observed quite high values of Se and lower values of Sp, thus confirming that the findings related to vascularization may mimic suspicious lesions subsequently ascertained as benign in nature [24].

The results of inter-observer agreement show a considerable difference between FFDM and US, MRI and CESM. The not negligible number of advanced breast cancers (T3 and T4) resulting in extended lesions observed on the mammogram may have significantly contributed to this result. Indeed, in extended cancers, differentiating one single large lesion from multiple small lesions can be particularly difficult. Results from the second-level breast imaging techniques with contrast media significantly add to the most appropriate evaluation of tumor extension.

Our study has some limitations. As mentioned, two experienced radiologists were involved in the evaluation of imaging studies. We did not evaluate the background parenchymal enhancement (BPE) in our study, which is of interest, as it is one of the main aspects that affects MRI and seems to be less relevant in CESM [25,26]. Along with MRI, CESM is influenced by BPE, which is a crucial aspect with which the radiologists must be familiar, in order not to miss malignant enhancement; however, while BPE is markedly influenced by menstrual cycle in MRI studies, it seems than this aspect is less relevant when performing CESM [25,26] (Figure 3). The results are not transferrable to a screening population, as only patients with unresolved findings at US and FFMD were included. An issue of further relevance relates to the undesired side effects related to the exposure to a non-ionic iodinated contrast agent (Visipaque 320) in women undergoing CESM. Indeed, CESM advantages have to be balanced against the risk of contrast agent reactions, which, although generally low and overall acceptable, are greater than those associated with gadolinium [3].

However, CESM may be an alternative for patient populations at high risk who are unable to undergo MRI [27], using its applications as a guide in performing biopsies and radiomics applications [28] in the near future. Of further note, computer-aided detection (CAD) and artificial intelligence (AI) were not included in the study design at this stage of our research. Nevertheless, we are aware of the importance of comparing and eventually integrating results from humans with those from expert systems. In future studies, CAD and AI will surely be among our operative tasks [29,30]. 

We would also cite in this context of critical discussion feasibility issues that emerged over the conduct of this trial, with potential implication in terms of bias in radiology. When considered globally, the performance of the totality of the imaging techniques included in our trial generated a washout interval of about 1 month, which may have consequences in terms of bias from image recall. Errors, discrepancy and bias in radiology represent highly debated research themes, with increasingly emerging evidence on the potential causes and consequences [31]. Further attention will be devoted to these tasks in future studies from our team.

## 5. Conclusions

Emerging evidence supports the potential role of radiomics in supporting clinicians and surgeons in decision making, though larger samples of patients need to be investigated [32]. We provide evidence in support of CESM in the clinical routine and with regard to the diagnostic workup of breast lesions. In this context, CESM may represent a valuable alternative and/or an integrating technique to MRI in the evaluation of breast cancer patients. Compared to MRI, CESM has the advantage of low cost, speed and accessibility, and being better tolerated by patients. Our results represent good-quality scientific evidence, obtained from an ad hoc conceived study, which was carried out according to a prospective design. Methodological aspects related to the study sample size were taken into account, resulting in an overall study population comparable to the other available studies. Nevertheless, further evidence is warranted to confirm our findings and address aspects related to imaging techniques’ comparability and accuracy in well-characterized subsets of patients with specific features.

## Figures and Tables

**Figure 1 cancers-14-01351-f001:**
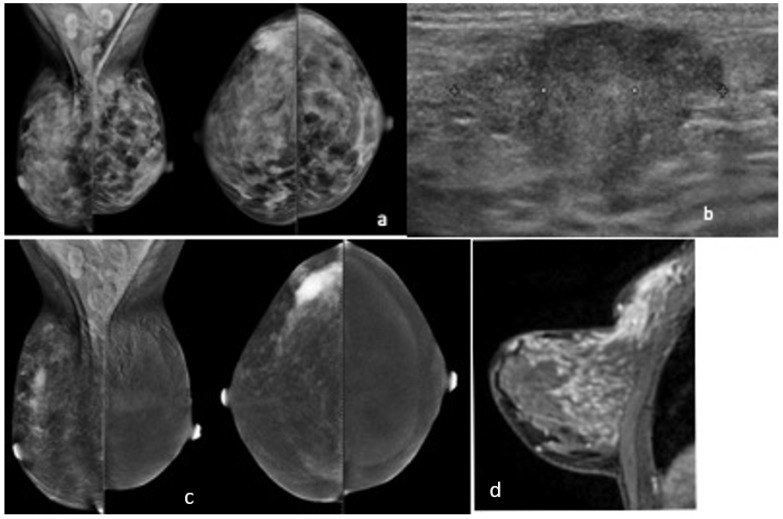
Local advanced breast cancer of the upper outer quadrant of the right breast; FFDM shows an area of hyperdensity (**a**); US confirmed a hypoechoic lesion (**b**), while MRI and CESM showed more extensive disease (**c**,**d**).

**Figure 2 cancers-14-01351-f002:**
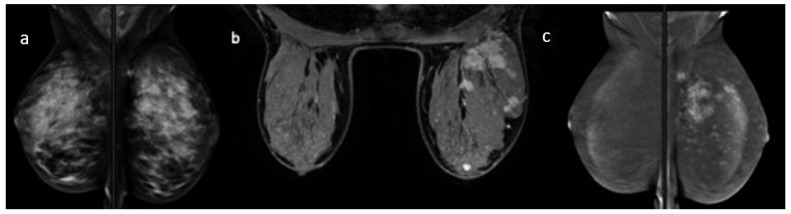
Local advanced breast cancer of the left breast. FFDM (**a**) does not allow for detection of the real disease extension of the lesion, in relation to breast density. MRI (**b**) and CESM (**c**) show multiple foci in relation to multicentric disease.

**Figure 3 cancers-14-01351-f003:**
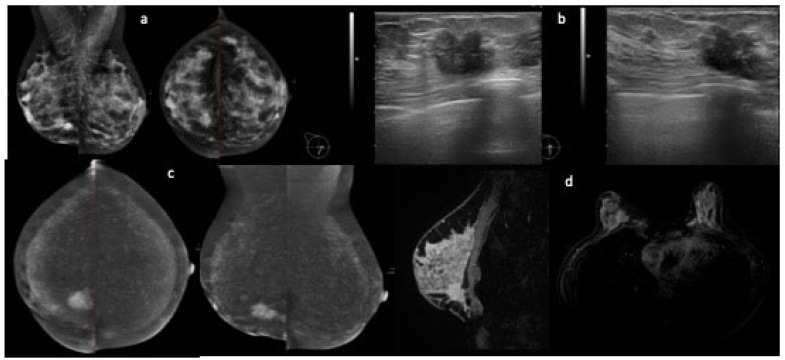
FFDM shows an irregular opacity in the inner lower quadrant of the right breast (**a**); US images show an irregular hypoechoic nodule in the inner lower quadrant of the right breast, with some suspicious smaller images close to the bigger one (**b**). CESM images confirm an irregular contrast uptake nodule in the inner lower quadrant of the right breast (**c**). Compared to the MRI, the background enhancement is still present but milder (**d**).

**Table 1 cancers-14-01351-t001:** Minimum (min), maximum (max) and mean dose exposure for FFDM and CESM.

	Dose Glandular Exposure	
	FFDM	CESM
Min	0.68 mGy	1.22 mGy
Max	1.32 mGy	2.28 mGy
Mean dose	1.04 mGy (±0.05)	1.75 mGy (±0.07)

**Table 2 cancers-14-01351-t002:** Characteristics of the study participants (*N* = 118).

Study Participants Caracteristics	*N* (%)
Age at study entry (years) (mean ± SD)	48.5 ± 9.7
Patients with multifocal breast cancer (*)	
No	70 (79.5)
Yes	18 (20.5)
Patients with bilateral breast lesions	
No	94 (79.7)
Yes	24 (20.3)
Bilateral breast cancer	
No	110 (93.2)
Yes	8 (6.8)
Multifocal and/or bilateral breast lesions	
No	81 (68.6)
Yes	37 (31.4)
Cytological/Histological assessment Right breast	*N* = 66 (55.9)
No cancer	27 (40.9)
Cancer	39 (59.1)
Cytological/Histological evaluation Left breast	*N* = 76 (64.4)
No cancer	17 (22.4)
Cancer	59 (77.6)

* Only cancer cases *N* = 88.

**Table 3 cancers-14-01351-t003:** Characteristics of the breast lesions by diagnostic test in 118 study participants.

BI-RADS	*N* (%)
FFDM Total lesions	105
Benign	1 (1.0)
Probably benign–suspect <2%	14 (13.3)
Low suspect <10%	21 (20.0)
Intermediate suspect 10–50%	24 (22.9)
High suspect 50–95%	12 (11.4)
Malignant >95%	33 (31.4)
US Total lesions	137
Benign	5 (3.6)
Probably benign–suspect <2%	31 (22.6)
Low suspect <10%	12 (8.8)
Intermediate suspect 10–50%	14 (10.2)
High suspect 50–95%	23 (16.8)
Malignant >95%	52 (38.0)
CESM Total lesions	110
Probably benign–suspect <2%	10 (9.1)
Low suspect <10%	10 (9.1)
Intermediate suspect 10–50%	17 (15.5)
High suspect 50–95%	27 (24.5)
Malignant >95%	46 (41.8)
MRI Total lesions	108
Benign	4 (3.7)
Probably benign–suspect <2%	13 (12.0)
Low suspect <10%	10 (9.3)
Intermediate suspect 10–50%	3 (2.8)
High suspect 50–95%	29 (26.8)
Malignant >95%	49 (45.4)

FFDM: full-field digital mammography; CESM: contrast-enhanced spectral mammography; US: ultrasound; MRI: magnetic resonance imaging.

**Table 4 cancers-14-01351-t004:** Concordance analysis between imaging techniques. *Per lesion* approach (*N*: 142).

	Overall		K Qualitative Scale	
	Gold Standard	CESM	<0.01	No Agreement
FFDM	0.69 *p* < 0.001	0.81 *p* < 0.001	0.01–0.20	Scarce
US	0.76 *p* < 0.001	_	0.21–0.40	Low
FFDM + US	0.68 *p* < 0.001	0.81 *p* < 0.001	0.41–0.60	Moderate
CESM	0.63 *p* < 0.001	_	0.61–0.80	High/Good
MRI	0.56 *p* < 0.001	0.73 *p* < 0.001	0.81–1.00	Excellent

**Table 5 cancers-14-01351-t005:** Concordance analysis between imaging techniques. *Per patient* approach (*N*: 118).

Right Breast	In NON-Multifocal and/or Bilateral Lesions		In Multifocal and/or Bilateral Lesions			
	Gold Standard	CESM	Gold Standard	CESM		
FFDM	0.70 *p* < 0.001	0.87 *p* < 0.001	1.00 *p* < 0.001	1.00 *p* < 0.001	K Qualitative Scale	
US	0.70 *p* < 0.001	_	0.74 *p* = 0.001	_	<0.01	No agreement
FFDM + US	0.61 *p* < 0.001	0.87 *p* < 0.001	0.74 *p* = 0.001	1.00 *p* < 0.001	0.01–0.20	Scarce
CESM	0.52 *p* = 0.003	_	0.77 *p* = 0.001	_	0.21–0.40	Low
MRI	0.37 *p* = 0.018	0.63 *p* = 0.001	0.44 *p* = 0.057	0.63 *p* = 0.011	0.41–0.60	Moderate
Left breast	In Non-Multifocal and/or bilateral lesions		In Multifocal and/or bilateral lesions		0.61–0.80	High/Good
	Gold standard	CESM	Gold standard	CESM	0.81–1.00	Excellent
FFDM	0.62 *p* < 0.001	0.64 *p* < 0.001	NA	NA		
US	0.78 *p* < 0,001	_	0.86 *p* < 0.001	_		
FFDM + US	0.64 *p* < 0.001	0.72 *p* < 0.001	0.83 *p* < 0.001	0.65 *p* = 0.001		
CESM	0.64 *p* < 0.001	_	0.64 *p* = 0.002	_		
MRI	0.71 *p* < 0.001	0.79 *p* < 0.001	1.00 *p* < 0.001	1.00 *p* < 0.001		

**Table 6 cancers-14-01351-t006:** Results for diagnostic accuracy (DA), sensitivity (SV), specificity (Sp), positive predictive value (PPV) and negative predictive value (VPN) for each modality and the gold standard.

		Gold Standard		Se	Sp	PPV	NPV	DA
		Positive	Negative					
FFDM	Positive	77	5	0.99	0.61	0.94	0.89	0.93
	Negative	1	8					
US	Positive	87	7	0.97	0.76	0.92	0.88	0.92
	Negative	3	22					
US + FFDM	Positive	91	11	0.98	0.63	0.89	0.90	0.89
	Negative	2	19					
MRI	Positive	74	10	0.99	0.47	0.88	0.9	0.88
	Negative	1	9					
CESM	Positive	87	7	1	0.5	0.92	1	0.93
	Negative	0	7					

**Table 7 cancers-14-01351-t007:** Results on the inter-observer agreement calculated on the findings of the two readers.

	KAPPA	*p*-Value	95% CI
FFDM	0.53	0.024	0.19–0.88
US	1.00	<0.001	1.00–1.00
MRI	1.00	<0.001	1.00–1.00
CESM	1.00	<0.001	1.00–1.00

## Data Availability

The data presented in this study are available on request from the corresponding authors.

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
