# Peer review of "Diagnostic Accuracy of Contrast-Enhanced, Spectral Mammography (CESM) and 3T Magnetic Resonance Compared to Full-Field Digital Mammography plus Ultrasound in Breast Lesions: Results of a (Pilot) Open-Label, Single-Centre Prospective Study"

_cancers, 2022, doi:10.3390/cancers14051351_

Round 1
Reviewer 1 Report
Dear authors,
This version of your paper missed the point. I don't see any improvements in points from the last time.
1. Introduction: goal, aims, contributions are missing.
2. "Written informed consent was secured from each participating patient." - What is the content of the consent? Is it approved by your ethical committee? Is there some proof of the truth or just a statement? Ethical issues are important here.
3. Image analysis was performed only by humans. It would be contribution to science to compare human results with some expert system or to develop such expert system. This would be a great contribution to the field. For example, micro calcifications can be detected by different color. Why the computer cannot detect tissues of interest in your case?
4. Section Statistical Analysis: Authors should discuss the meaning of their statistics in manner interesting to the readers.
5. The receiver operator characteristics curve is missing to evaluate accuracy of the diagnostic procedures to predict the nature and characteristics of the examined breast lesions. It would be interesting to further elaborate ROC. And, if possible, to present some figure of ROC in your case. And math expression of it. Instead of deep study of the subject, you deleted this issue.
Author Response
Please find the response in the attachment

Reviewer 2 Report
You manuscript is very well-written. The methodological aspects are well designed. It also has very good summary on the advantages and limits of diagnostic methods including CESM and MRI.
There are only two minor issues in the manuscript:
- page 19, line 327: CESM may provide “an” (but not "and") effective tool in the evaluation of tomosynthesis-detected architectural distortion (AD).
- Page 3, line 81, the format has a very minor issue.
Considering the significance and the quality of this study, I recommend acceptance with only minor revisions. We look forward to your revised version for publishing!
Thanks
Author Response

(The authors gave the same response as above.)

Round 2
Reviewer 1 Report
Answers are satisfactory.
This manuscript is a resubmission of an earlier submission. The following is a list of the peer review reports and author responses from that submission.
Round 1
Reviewer 1 Report
Specific Comments
- figure 3: It can be definetly improved. Also, in the caption you should use the BI-RADS lexicon. "impetuus" is not in the BI-RADS lexicon
- reference 17 is spelled differently compared to the others. Please amend.
Reviewer 2 Report
The current study is on a topic of relevance and general interest to the readers of the journal. I found the paper to be overall well written and felt confident that the authors performed careful and thorough research in support of CESM in the clinical routine and integrating with MRI in the evaluation of breast cancer patients.
I had the following suggestions
1) Did the authors think of tomosynthesis and how this may be effective in terms of using with CESM ?
2) summarize the challenges that the use of CESM may be restricted ; the authors do mention the opportunities as in high risk
Reviewer 3 Report
"Diagnostic Accuracy of Contrast-enhanced, Spectral Mammography (CESM) and 1 3T Magnetic Resonance Compared to Full Field Digital Mammography" - from title - it is not shown in the paper. There are not scientific methodology for comparison of stated. It is not compared in proper manner. You should have parallel data of two or more instruments. For example, classic and 3T MR parallel comparison of the results for the same patients.
179-181 "for the same patient, the first reading session included images from FFDM + US and 3T MRI, whereas CESM-related images were evaluated in the second reading session ". Is there a reason for such order? Why didn't you choose reverse order of readings?
Did you have a ground truth (what is actually correct) or you just compare two readers? You should compare human readers to some ground truth.
References
You could consider:
https://doi.org/10.1016/j.breast.2020.06.005
https://doi.org/10.1590/0100-3984.2016-0029
https://dx.doi.org/10.1371%2Fjournal.pone.0190287
https://www.zgt.nl/media/15961/multireader-study-on-the-diagnostic-accuracy-of-ultrafast-breast-magnetic-resonance-imaging-for-breast-cancer-screening.pdf
https://doi.org/10.4236/ojcd.2019.91003